# ASK2MASK: GUIDED DATA SELECTION FOR MASKED SPEECH MODELING

## ABSTRACT

Masked speech modeling (MSM) methods such as wav2vec2 or w2v-BERT learn representations over speech frames which are randomly masked within an utterance. While these methods improve performance of Automatic Speech Recognition (ASR) systems, they have one major limitation. They treat all unsupervised speech samples with equal weight, which hinders learning as not all samples have relevant information to learn meaningful representations. In this work, we address this limitation. We propose ask2mask (ATM), a novel approach to focus on specific samples during MSM pre-training. ATM employs an external ASR model or *scorer* to weight unsupervised input samples in two different ways: 1) A fine-grained data selection is performed by masking over the highly confident input frames as chosen by the scorer. This allows the model to learn meaningful representations. 2) ATM is further extended to focus at utterance-level by weighting the final MSM loss with the utterance-level confidence score. We conduct fine-tuning experiments on two well-benchmarked corpora: LibriSpeech (matching the pre-training data) and Commonvoice, TED-LIUM, AMI and CHiME-6 (not matching the pre-training data). The results substantiate the efficacy of ATM on significantly improving the recognition performance under mismatched conditions (up to 11.6% relative over published results and upto 4.46% relative over our internal baseline) while still yielding modest improvements under matched conditions.

## 1 INTRODUCTION

Self-training and self-supervised training techniques rely on huge amounts of unlabeled speech or text data for better generalization. The self-training techniques such as pseudo-labeling (Scudder, 1965; Kahn et al., 2020) and student-teacher training (Park et al., 2020) have shown promising improvements by incorporating the data selection process. This data selection step removes pseudo-labels with less confidence as denoted by the teacher model before feeding the input to a student model. Xu et al. (2021) shows that self-training and self-supervised training are complementary to each other and also show that self-supervised models act as good initialization for self-training techniques. Self-supervised training (Hinton & Zemel, 1994) is a representation learning approach which implicitly learns patterns in the data without relying on explicit labels. Masked speech modeling (MSM) is the recent and successful self-supervised learning technique, thanks to the advent of BERT (Devlin et al., 2018) in NLP which inspired learning speech representations from masked inputs. MSM techniques such as wav2vec2 (Baevski et al., 2020), HuBERT (Hsu et al., 2021b) and w2v-BERT (Chung et al., 2021) have shown considerable gains across various down-stream speech tasks and have become the go-to models for ASR.

Unfortunately, MSM does not have a data selection scheme to discard the irrelevant input samples and instead imposes burden on the training criterion to learn the relevance of the input samples in learning meaningful representations. Hsu et al. (2021a) noticed the impact of not selecting relevant data from the huge amounts of unsupervised data during pre-training by showing degradation in ASR performance when fine-tuned to a target dataset with limited data. To mitigate this constraint, Chan et al. (2021) introduced substantially more fine-tuning data related to the target dataset but did not achieve satisfactory results. Hsu et al. (2021a) attempted to solve this issue by heuristically selecting the data from a closed set of unsupervised speech databases or by pooling in data relevant to target dataset along with the existing pre-training dataset. However, this data selection approach is not done within the existing pre-training dataset and it is not completely empirically motivated.

In this study, in order to break the above limitation of the MSM techniques, we propose a simple strategy named *ask2mask (ATM)* to incorporate data selection within a chosen pretraining dataset.

- In ATM, the masking is done over the input samples or speech frames with higher confidence as determined by the scorer. This is contrary to the random selection of frames to be masked in conventional MSM models. We hypothesize that this guided selection of frames to be masked allows the model to focus on the frames which can provide meaningful representations. The scoring model used in this work is necessarily a speech recognition model trained on small amount of data and provides frame-level confidence for each input.

- The ATM technique is further extended to exploit the confidence values provided by the scorer by directly using them to re-weight the MSM loss. We denote this approach as ATM with loss scaling (ATM+S). It allows the model training to focus on certain *utterances* by down scaling the utterances with low-confidences.

Similar to our work based on masking with external guidance, there is work in NLP that also benefit by incorporating masking with knowledge. In Sun et al. (2019), masking is done at phrase-level segments in BERT and has shown to learn semantic dependencies. In Wang et al. (2019), phonetic knowledge is injected to mask over phonetic segments to perform spectral augmentation.

Our ATM approach is primarily motivated based on the recent work by Veselý et al. (2017) on semi-supervised learning of conventional ASR systems which shows that performing data selection at frame-level or token-level on unsupervised data provides better performance. The importance of pruning out the input samples at frame-level has been studied in Ferreira et al. (2021) to improve both classification and regression tasks. Few works on unsupervised learning also highlight the importance of weighting the data based on its confidence (Wessel et al., 2001; Ren et al., 2020; Coleman et al., 2019). We hypothesize that ATM can leverage the effect of data selection within a particular training corpus to further enhance the recognition performance of MSM techniques.

To summarize, our contributions are listed as follows:

- *Novelty*: To the extent of our knowledge, ATM is the first approach to incorporate a within-corpus data selection strategy in MSM. We also show that data selection can be simply performed inside MSM by guided selection of frames to be masked using a scorer model.

- *Technical contributions*: We provide two simple strategies to incorporate data selection into MSM pretraining by applying the confidence of the scorer: 1) choosing the data at frame-level by applying guided masking 2) soft weighting the data at utterance level by scaling the MSM loss of each utterance with its corresponding confidence score. ATM is designed to be compatible to all MSM based pre-training techniques.

- *Empirical study*: Analysis is done to find an optimal masking percentage for ATM and we highlight the effectiveness of ATM across varying masking percentages. The importance of masking frames with high confidence is substantiated by empirically comparing it with masking low confident frames and random frames respectively. Experiments are performed on AMI data which is from a distinct condition compared to Libri-light corpus used for MSM based pretraining. The results confirm the importance of ATM by improving the recognition performance on evaluation sets of AMI by a significant margin.

## 2  PRELIMINARIES ON MASKED SPEECH MODELING (MSM)

In this section, we formally define the masked speech modeling (MSM) technique and brief primary instantiations including wav2vec2 and w2v-BERT. The technique can be formulated by defining input speech sequence $\mathbf{X} = [x_1, x_2, ..., x_{T'}]$, where $x_t$ is the log Mel-filterbank feature vector at time $t$. $\mathbf{X}$ is sent to the feature encoder $\Phi$ to obtain the encoded representations $\mathbf{E} = \Phi(\mathbf{X})$. The feature encoder contains convolutional layers performing subsampling at a factor of 4 and reducing the total number of frames of an utterance from $T'$ to $T$ to get $\mathbf{E} = [e_1, e_2, ..., e_T]$. $\mathbf{E}$ is then sent to two parallel modules: 1) masking component, and 2) quantizer.

## 2.1 MASKING

The idea behind masking input samples and predicting them was initially proposed in BERT (Devlin et al., 2018) and later adopted to speech (Baevski et al., 2020) with modifications to suit the characteristics of speech input. The masking is done over sets of frames or blocks $b_1, b_2, ..., b_K$ and accommodates overlap between blocks. Here $K$ is the number of masked blocks in a randomly masked encoded sequence $\tilde{\mathbf{E}}$. The importance of block masking is motivated by the improvements observed in Span-BERT by Joshi et al. (2020) and ERNIE (Sun et al., 2019). The block $b_k = [i_k, c]$, where $i_k$ is the starting index of the masked block and $c$ is the corresponding right context size denoting the number of consecutive speech frames. Here $i_k$ are randomly sampled from a uniform distribution. It has been empirically observed by Baevski et al. (2020) that 49% of the frames are masked and $c = 10$ is chosen as the golden ratio to attain best representation during pre-training.

## 2.2 QUANTIZER

Gumbel-softmax quantizer component $\Psi$ is used to get quantized representations which act as targets for wav2vec2 and w2v-BERT models. These quantized representations align to phonetic units as described in Baevski et al. (2020). Each quantized vector is of $L$ dimensions which denote the number of targets or codes used in a codebook. Each incoming input $\mathbf{E}$ is projected to $L$ dimensions within the quantizer before applying the Gumbel-softmax.

## 2.3 CONTEXT NETWORK AND MSM LOSS

*Wav2vec2-conformer*: In this model type, the unmasked sequence $\mathbf{E}$ is sent to $\Psi$ to get $\mathbf{Q} = \Psi(\mathbf{E})$, where $\mathbf{Q} = [q_1, q_2, .., q_T]$ as described in Baevski et al. (2020). The masked sequence $\tilde{\mathbf{E}}$ is fed to the context network $\Omega$ which contains conformer blocks to learn contextual representations from the input. $\mathbf{C} = \Omega(\mathbf{E})$ denotes the output of the context network. The contrastive loss $\mathcal{L}_{ctr}(c_j, q_j)$ objective is computed between the quantized representation $q_j$ and context network output $c_j \in \mathbf{C}$ for all masked time instances $j \in J$. Diversity loss $\mathcal{L}_{div}$ is computed as an auxiliary objective in wav2vec2 to force the model to choose diverse codes in the quantization codebook. Detailed description of $\mathcal{L}_{div}$ is in Baevski et al. (2020). The final training objective is denoted as:

$$\mathcal{L}_{wv} = \mathcal{L}_{ctr} + 0.1 \cdot \mathcal{L}_{div}, \tag{1}$$

where $\mathcal{L}_{ctr} = \sum_{j=1}^{J} \mathcal{L}_{ctr}(c_j, q_j)$.

*HuBERT-conformer*: This is another variant of wav2vec2-conformer model with two major differences: 1) Targets are k-means cluster ids which are computed over a small portion of input 2) Cross-entropy loss $\mathcal{L}_{ce}(\hat{y}_j, y_j)$ is computed between the prediction of the context network $\hat{y}_j$ and the k-means cluster id target $y_j$.

*W2V-BERT*: This model marries the concept of wav2vec2 and BERT model by including an additional context network $\Lambda$ containing conformer blocks in addition to $\Omega$ as in wav2vec2. The $\Lambda$ receives the output of the $\Omega$ and strives to further learn refined contextual information to get $\mathbf{H} = \Lambda(\mathbf{C})$. The targets of w2v-BERT $y_j$ is computed by taking an argmax over the codebook dimensions $L$ of quantized representations $q_{j,l}$ as:

$$y_j = \arg\max_l q_{j, l}, l \in L \tag{2}$$

Finally, the cross-entropy loss $\mathcal{L}_{ce}(\hat{y}_j, y_j)$ is computed between the prediction $\hat{y}_j = \text{softmax}(h_j)$ and the target $y_j$ over the masked time instances $J$. The final training objective $\mathcal{L}_{wb} = \mathcal{L}_{ce} + \mathcal{L}_{wv}$ is a combination of cross-entropy loss and wav2vec2 loss. A block diagrammatic overview of the above MSM architectures are available in appendix A.7.

## 3 ASK2MASK (ATM)

The primary reason to employ pre-training models is to exploit the abundantly available unsupervised data for improving ASR under limited availability of supervised data. While the MSM models such as wav2vec2 and w2v-BERT described in Section 2 exploit the unsupervised data, they treat

each data with equal weight for computing the final objective.Instead, we generate a score $s_t$ for each encoded frame $e_t$. This is used to select relevant data in a fine-grained manner during masking for computing the loss objective.

## 3.1 METHODOLOGY

For each encoded feature frame $e_t \in \mathbf{E}$, the scorer emits probabilities $p(v_t = l \,|\, \mathbf{E}); l \in \mathbf{L}$ of the frame belonging to a particular label. The scorer model is a CTC based frame-synchronous ASR model separately trained with a limited amount of data. Our initial intuition was to chose the scorer's training data to match the target data condition, however our empirical analysis in (cf. Section 5.3) shows that the performance is agnostic to the scorer model's training data. Finally, the confidence score $s_t$ of the frame is defined as the maximum probability across all labels:

$$s_t = \max_l p(v_t = l \,|\, \mathbf{E}) \tag{3}$$

We sample $K$ masking start indices $\{i_1, .., i_k\}$ with probabilities:

$$p(i_k = t) = \frac{s_t}{\displaystyle\sum_{v \notin \{i_1,..,i_{k-1}\}} s_v} \cdot \delta_{t \notin \{i_1.,,i_{k-1}\}}, \tag{4}$$

That is, we sample beginning frames with probability proportional to the scores of each frame. The indicator function $\delta_{t \notin \{i_1.,,i_{k-1}\}}$ ensures that we sample *without* replacement. This is the key difference between ATM and the random masking in prior works as described in Section 2.1. Prior works uniformly sample the start indices of each masking block $b_{1:K}$, while the ATM uses the probability distribution induced by the scorer. $K$ is determined by the percentage of frames to be masked.

We hypothesize that frames with maximum confidence from an external scoring model will be 1) easiest to learn using an MSM training criteria and 2) most informative in for pretraining to facilitate fine-tuning. Conversely, the lowest confidence frames, those more confusable to an external scoring model, will be the least reliably learned by MSM and least informative for pretraining.

The resulting frames are sent as input to the MSM architecture and the final loss objective $\mathcal{L}$ is determined by either of the MSM objectives $\mathcal{L}_{wv}$ or $\mathcal{L}_{wb}$ described in Sections 2.3. This modified training objective allows the model to focus on learning from gradients calculated from the frames with high confidences.

## 3.2 ATM WITH MSM LOSS SCALING (ATM+S)

The ATM loss is computed over the frames with high confidence performing a fine-grained data selection within a $u^{\text{th}}$ speech sequence $\mathbf{X}_u$. Utterances with higher average frame confidence as measured by the scorer are accorded higher value than those with more confused frames. To perform data selection at a coarser utterance level, confidence scores $s_u$ are computed as:

$$s_u = \frac{1}{T} \sum_{t=1}^{T} s_{t,u} \tag{5}$$

For simplicity, we denote $s = s_u$ and the MSM loss computed over each masked frame is scaled by $s$ to impose the importance of a particular utterance $u$. The final training objective $\mathcal{L}'_{atm}$ of a particular speech sequence is denoted as:

$$\mathcal{L}'_{atm} = s \cdot \mathcal{L}_{atm} \tag{6}$$

## 3.3 PROBABILITY AS CONFIDENCE MEASURE IN ATM

The ATM uses probability as a simple form of confidence measure to each frame. The analysis of confidence measures for semi-supervision in ASR has been done in Veselý et al. (2017) and they show that posterior probability acts as a reliable confidence measure for frame, word and sentence

based data selection. They also perform an extensive analysis on using the posteriors for hybrid ASR systems. Based on the motivation from this work we chose to use softmax probability directly as our confidence score. A similar observation has been noted in Ferreira et al. (2021), where the usage of softmax probability directly as a confidence measure has been applied to select relevant data samples during training. We also experimented with Entropy and exponential scaling or log scaling on softmax probabilities as confidence measure, but it did not fetch advantage over simple usage of probability.

## 4 EXPERIMENTAL SETUP

All experiments including pre-training and fine-tuning are performed using 80 dimensional log Mel-filterbank features computed over the sampled 16kHz audio. Datasets (such as AMI) contains wide-band audio and are downsampled to 16kHz. We evaluate with the test-other (LibriSpeech partition) to show the importance of ATM on matched data conditions, while IHM-eval and SDM-eval (AMI partitions) is used to validate the model under mismatched conditions.

### 4.1 DATASETS USED

*Pretraining (PT)*: Libri-light (LL-60k) dataset contains 60k hours of unlabeled speech and is used to pre-train all MSM models. LL-60k is the most widely used large unsupervised speech corpus for various PT techniques. Each input speech sequence is constructed by first randomly selecting 32-64 seconds segments from the original utterance. From these segments, a contiguous 32 second region is extracted from a random starting point on-the-fly during MSM PT as described in Zhang et al. (2020b).

*Finetuning (FT)*: Different target datasets including 1) 100 & 960 hours of Librispeech (LS-100 & LS-960) (Panayotov et al., 2015). 2) 100 hours of AMI and 3) speechstew (approx. 5k hours) (Chan et al., 2021) are used to perform our FT experiments. Each dataset used is specific to a certain target data condition, for instance LS-960 is closely matches the LL-60k, AMI dataset is distinct from the LL-60k condition and it contains speech from two kinds of microphones (i) Independent head microphone (IHM). (ii) single distant microphone (SDM). SpeechStew is composed of datasets chosen from multiple conditions to create a mixed domain aggregate corpus. Details of its processing are described in Chan et al. (2021).

*Evaluation*: We hypothesize that evaluation over AMI using IHM-eval and SDM-eval reveals the effectiveness of ATM in providing informative samples for better representation learning. We also evaluate using evaluation sets from Tedlium and Common voice as their training counter parts are used in SpeechStew based FT. Finally, we also evaluate using CHiME-6 Watanabe et al. (2020) without using any FT data from CHiME-6 training set to compare the performance of ATM on completely unseen target dataset.

*Scorer training data*: A CTC (Graves et al., 2006) based conformer model with 100M parameters is trained on LS-100 ("LS-scorer"). A similar model is also trained on AMI ("AMI-scorer"). Word-piece model (WPM) with 1024 tokens are used as labels for training the scorer models. All the results in this paper use "LS-scorer" besides the comparison Section 5.3.

### 4.2 MSM ARCHITECTURE

*Wav2vec2-conformer*: This is a wav2vec2 with conformer based context network which first encodes the filterbank features using two 2D convolutional layers with strides (2,2). Model has 100M/600M parameters and is denoted as "w2v2-conformer-L/XL". HuBERT-conformer-L/XL is similar to w2v2-conformer-L/XL - it differs in using the k-means based quantizer with 1024 targets and computes the cross-entropy loss as described in Hsu et al. (2021b). The "L/XL" size models contains context network $\Omega$ 12/24 conformer layers with 8 attention heads and 1024 hidden dimensions.

*W2v-BERT*: W2v-BERT is explored using two model sizes: one with 100M parameters denoted as "w2v-BERT-L" and containing 2 conformer layers in context net $\Omega$ and 4 conformer layers in $\Lambda$. A 600M parameter model is denoted as "w2v-BERT-XL" contains 8 conformer layers in $\Omega$ and 24 conformer layers in $\Lambda$. Each conformer block contains 1024 hidden dimensions with 8 attention

heads, kernel size of 5 with local context of 128. The remaining architecture is identical to the configuration defined in Chung et al. (2021).

### 4.3 PT AND FT CONFIGURATION

The models L/XL are trained with a global batch size of 512/2048 on 64/256 Google TPU V3 cores for 2-4 days respectively. Adam optimizer is used with a learning rate schedule (Section 5.3 of Vaswani et al. (2017)) with 2e-3 as peak learning rate and 25k warmup steps. The model training configuration follows similar procedure as described in Zhang et al. (2020b).

The FT is done by employing the context network from the PT model by adding a final layer with 1024 WPM units learnt using the RNN-T objective function (Zhang et al., 2020a). The FT is done on w2v-BERT-XL, w2v2-conformer-XL and HuBERT-conformer-XL after 400k PT model updates. The w2v-BERT-L model is FT after 900k PT model updates. w2v-BERT-L is used to initially perform wide range of analysis and hyper-parameter optimization on ATM. w2v-BERT-XL is finally used to compare the results of ATM across existing works in literature. w2v2-conformer-XL and HuBERT-conformer-XL are also used in our experiments. All these models are trained with the same configuration as in Zhang et al. (2020b).

## 5 ATM ANALYSIS

The empirical study on ATM is done primarily using w2v-BERT-L since this generates the best WER performance across similarly sized models (cf. Figure 2). The pre-trained models are fine-tuned with either LS-100 or AMI. The resulting finetuned models are evaluated on IHM and SDM evaluation sets to understand the domain generalization aspect of ATM. Librispeech evaluation sets are used in unison to study how ATM behaves under matching domain condition. These experiments are performed with using the loss scaling (it will be discussed in Section 5.5).

### 5.1 MASKING PERCENTAGES

The number of masked frames within an utterance plays a key role in masked input learning and in this study, we vary the masking percentages from 30% to 50% to determine the best percentage for ATM approach. Previous works on wav2vec2 (Baevski et al. (2020)) showed that masking 49% of the frames is ideal for 30 second utterance and this has been followed subsequent works such as HuBERT and w2v-BERT. In case of ATM, this can differ as the frames selected are of higher confidences. Figure 1 shows that ATM achieves its "sweet spot" with 40% masking for both IHM-eval and test-other set. Interestingly ATM's performance is stable across large variations in masking rates with relatively good performance with masking rate as low as 30%. This is a significant difference from the uniform sampling of prior work which suffers significant drop in performance as the masking rate goes below 40%. The result indicates that masking the right set of frames, which ATM aims to do, is able to promote more stable performance. For instance, ATM achieves a %WER of 12.65 with 33% masking and 12.52 with 40% masking on IHM-eval respectively as shown in Figure 1. The

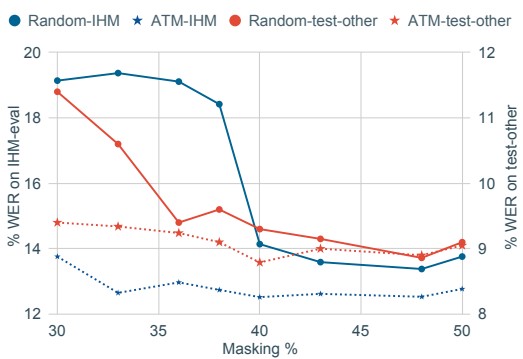

Figure 1: Recognition performance of w2v-BERT with ATM and random masking on IHM-eval and test-other sets by varying the masking percentage during pre-training. The FT is performed on LS-100 for evaluating test-other, while IHM-eval is evaluated with model FT with AMI. Random masking shows a substantial shift in performance when varying the masking from 30% to 40%, while the ATM remains robust to changes in masking percentage.

recognition performance on test-other and IHM-eval improves over baseline from 8.86% to 8.79% and 13.38% to 12.52% respectively by using ATM.

## 5.2 ATM MASKING STRATEGIES

The default setting of ATM is chosen based on a hypothesis that those frames that are scored with high-confidence from an external scoring model will be most useful as candidates for MSM pretraining. This hypothesis is interrogated in this section by analysing the impact of choosing the frames with low confidences or equal mix of both high and low confidence frames (Mixed). For masking low confident frames, we modify the score in Equation 3 as:

$$s_t = 1 - \max_l p(v_t = l \,|\, \mathbf{E}) \qquad (7)$$

Table 1: Performance comparison (in %WER) on AMI evaluation sets using w2-BERT with random masking (baseline) and with ATM using high, low and mixed confidence scores from the scorer. The FT is done with AMI.

| Model | Confidence Level | PT-LL, FT-AMI | |
| --- | --- | --- | --- |
| | | IHM-eval | SDM-eval |
| Baseline | Random | 13.38 | 31.63 |
| ATM | High | 12.52 | 27.34 |
| | Low | 14.14 | 37.77 |
| | Mixed | 13.96 | 30.51 |

We evaluated these three masking strategies of ATM on both IHM and SDM evaluation sets. Table 1 shows the comparison between these sampling strategies. We observe that masking high confident frames are consistently better than masking the low confidence counterparts. In fact, "Low" confident frames perform worse than the baseline with random masking. Finally, we observe that performance of "Mixed" falls between that of "High" and "Low". The "Mixed" strategy is similar to random masking, as both high and low confidence frames are selected. This similarity is also reflected in comparable performance between "Mixed" and random masking. These results provide support for our initial hypothesis that masking frames with high confidence leads to better pre-training.

## 5.3 HOW TO CHOOSE THE SCORING MODEL

The scorer used in this work is a speech recognition model (100M parameters) trained in a supervised fashion. The scorer is chosen based on the target downstream task and in addition to this, the scorer needs to be frame-synchronous to provide confidence for every frame in a speech sequence. In this work, we use a frame-synchronous ASR system as the scorer by employing the connectionist temporal classification (CTC) objective. The CTC is preferred over the RNN-T by analysing the reliability of the frame-level predictions. To analyse the importance of the supervised data used to train the scorer, we trained two scorer models: LS-scorer and AMI-scorer are CTC models trained with LS-100 and AMI dataset respectively. The AMI-scorer outperforms on SDM-eval

Table 2: Cross analysis of ATM performance (in %WER) using AMI and LS scorers. The FT is done on LS-100 to evaluate the test and test-other, while the FT is done on AMI to evaluate using IHM-eval and SDM-eval

| Evalset | LS-scorer | AMI-scorer |
| --- | --- | --- |
| test | 3.89 | 3.93 |
| test-other | 8.92 | 9.68 |
| IHM-eval | 12.52 | 12.3 |
| SDM-eval | 27.34 | 27.00 |

by improving the %WER from 27.34 to 27.00. Surprisingly, our results on Table 2, shows that the results on IHM-eval using an LS-scorer are comparable to the AMI-scorer. Evaluation on test and test-other shows that LS-scorer is better than AMI-scorer on both sets. Based on these observations, we choose the LS-scorer as the universal scoring model for all ATM based pre-trained models regardless of the target domain (eg: AMI) used in our experiments. Table 2 shows that although matching the scorer to the target domain improves the performance, the difference is not significant.

## 5.4 CONSISTENCY ACROSS DIFFERENT ARCHITECTURES

Figure 2 shows that ATM consistently outperforms on both IHM-eval and SDM-eval across multiple MSM architectures including wav2vec2 and HuBERT. In the case of IHM-eval, ATM attains a relative improvement of 9% over w2v2-conformer-L, 4% relative improvement over HuBERT-conformer-L and 5% relative gain over w2v-BERT-L baseline models respectively. W2v2-conformer-L using ATM obtained 6.2% relative improvement over its baseline counterpart and HuBERT-conformer-L with ATM attained 7.9% rel. improvement over HuBERT-L baseline on SDM-eval respectively. On the other hand w2v-BERT-L baseline is better compared to w2v2-conformer-L and HuBERT-conformer-L on both IHM-eval and SDM-eval by achieving 12.52% and 27.34% WER respectively.

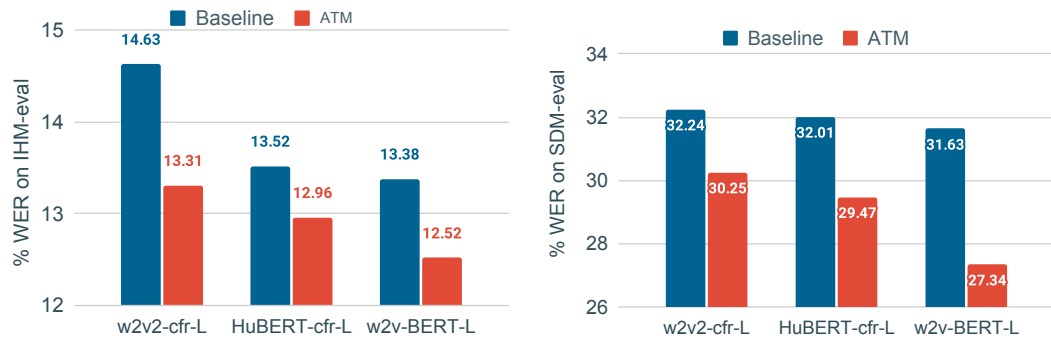

Figure 2: Performance comparison of different MSM architectures with and without applying ATM on IHM-eval and SDM-eval in AMI. All these models are FT using AMI. Here "cfr" refers to conformer.

## 5.5 ATM WITH LOSS SCALING (ATM+S) ANALYSIS

ATM training can incorporate utterance-level weighting by scaling the MSM loss obtained using w2v-BERT-L models with the utterance-level confidence score according to Equation 6. We evalu-

Table 3: %WER of ATM+S by fine-tuning on AMI using w2v-BERT-L model

| MSM arch. | Type | IHM-eval | SDM-eval |
|-----------|------|----------|----------|
| w2v-BERT-L | Baseline | 13.38 | 31.63 |
| | Baseline+S | 13.14 | 28.16 |
| | ATM | 12.52 | 27.34 |
| | ATM+S | 13.05 | 27.19 |

ate the value of utterance-level loss scaling by re-weighting utterances in the context of both baseline MSM (i.e., without ATM frame selection) and ATM (ATM+S). These results are in Table 3. Re-weighting utterances by scaling the MSM loss with the confidence score on baseline model is denoted as "Baseline+S" and on ATM is labeled as "ATM+S". MSM loss scaling is effective even without ATM; baseline+S improves over baseline on both IHM-eval and SDM-eval. Moreover, ATM+S improves over ATM on SDM-eval by attaining 27.19% WER while showing degradation on IHM-eval. This shows that ATM+S is effective on very hard evaluation task such as SDM-eval compared to IHM-eval. On the IHM-eval test set, the impact of MSM loss scaling is observed over the Baseline MSM without ATM. We hypothesize that ATM+S may not able to provide improvement on IHM-eval as ATM already incorporates optimally incorporates scorer information on this task.

## 6 RESULTS

In this Section, XL models are used to compare the importance of ATM on LS-960, AMI and SpeechStew. These three datasets show the effect of ATM on diverse conditions with a much larger model. Results are compared with appropriate prior work. Table 4 shows that ATM and ATM+S improves over dev-other while on dev set there was improvement only using ATM and not ATM+S. Although the ATM and ATM+S does not show improvement on test and test-other, matches the very strong baseline. Considering the similarity between LS-960 and PT data, ATM manages to provide gains without hurting the performance across all Librispeech evaluation sets. This validates our argument that MSM models are better without any data selection when trained under matched data condition but can benefit under mismatched conditions.

Table 5 presents the results of ATM on AMI by comparing it with w2v2-conformer-XL baseline and w2v-BERT-XL baselines. We include w2v2-conformer-XL to further test the consistency of ATM on XL models when evaluated on harder tasks. ATM+S and ATM observes consistent gains over baseline on both IHM-eval and SDM-eval when trained with XL models. However, ATM+S did not

Table 4: %WER obtained by FT with LS-960 using w2v-BERT-XL model using baseline, ATM and ATM+S. The results show the impact of our proposed approach on matched condition since Librispeech evaluation sets are treated as closer to Libri-light PT domain.

| MSM arch. | Type | dev | dev-other | test | test-other |
|---|---|---|---|---|---|
| w2v2-conformer-XL | Baseline (Chung et al., 2021) | 1.7 | 3.5 | 1.7 | 3.5 |
| w2v-BERT-XL | Baseline (Chung et al., 2021) | 1.5 | 2.9 | 1.5 | 2.9 |
| | ATM | 1.4 | 2.8 | 1.5 | 2.9 |
| | ATM+S | 1.5 | 2.8 | 1.5 | 2.9 |

Table 5: %WER obtained by FT with AMI using w2v-BERT-XL model using baseline, ATM and ATM+S. Evaluation is done on AMI test sets to highlight the effect on mismatched condition.

| MSM arch. | Type | IHM-eval | SDM-eval |
|---|---|---|---|
| w2v2-conformer-XL | Baseline | 10.4 | 25.7 |
| | ATM | 10.0 | 24.5 |
| | ATM+S | 9.8 | 23.9 |
| w2v-BERT-XL | Baseline | 10.1 | 25.1 |
| | ATM | 9.5 | 23.7 |
| | ATM+S | 9.5 | 23.5 |

demonstrate improvement on IHM-eval using w2v-BERT-XL. Table 6 analyses the effect of ATM

Table 6: Comparison with state-of-the-art results on SpeechStew. The FT is done on SpeechStew and the results are evaluated using Kaldi scoring to match published results. Note that the model has *never* seen any CHiME-6 data, and we use it as an example for **zero-shot** learning mode.

| Model | Type | Commonvoice | Tedlium | AMI | | CHiME-6 |
| | | | | IHM | SDM | |
|---|---|---|---|---|---|---|
| Speechstew (Chan et al., 2021) | − | 12.1 | 5.3 | 9.0 | 21.7 | 57.2 |
| w2v2-conformer-XL (Chan et al., 2021) | − | 11.5 | 5.6 | 9.6 | 23.8 | 56.4 |
| w2v-BERT-XL | Baseline | 11.2 | 5.3 | 9.2 | 21.5 | 55.5 |
| | ATM | 10.8 | 5.3 | 9.0 | 21.0 | 54.3 |
| | ATM+S | 10.7 | 5.2 | 8.9 | 20.7 | 53.9 |

and ATM+S on multiple evaluation sets such as Commonvoice, Tedlium, AMI and CHiME-6. These four sets are chosen based on the mismatch range from minimum to maximum and for instance, Commonvoice has the minimum mismatch with Libri-light data, while CHiME-6 has the maximum mismatch. The state-of-the-art results published in Chan et al. (2021) are obtained by choosing the best Conformer model supervisedly trained with multiple datasets such as AMI, CommonVoice, Broadcast News, Librispeech, Switchboard/Fischer, TED-LIUM and Wall Street Journal. Note that the training data did not include the CHiME-6 data. The authors in Chan et al. (2021) show that simply training an ASR with lots of data leads to best results compared to the wav2vec2 finetuned model. Their best results are denoted in table 6 and will be used to compare with our best ATM results.

Our baseline w2v-BERT-XL attained better results over the published w2v2-conformer-XL and Speechstew results. In Commonvoice and CHiME-6, the baseline attained 7.4% and 2.9% relative improvement over Speechstew respectively. However, by including our ATM and ATM+S with w2v-BERT-XL, there was consistent improvement across all range of mismatched domains. For instance, ATM+S attains 5.76% relative improvement on CHiME-6 over the Speechstew. This result clearly justifies that selection of reasonable input samples during pre-training reduces the necessity of having finetuning data from the same domain to improve performance. To further substantiate this, the results on AMI show a 4.6% relative improvement on AMI-SDM over Speechstew which is of different domain compared to pre-training domain. In case of minimal mismatch domain such as

Commonvoice, the ATM attained 11.6% relative improvement over Speechstew. These observations show that ATM and ATM+S demonstrate their effectiveness to generalize to unseen and challenging speech recognition conditions.

## 7 CONCLUSION

In this work, we introduce ask2mask (ATM) to perform data selection over unsupervised samples for MSM based pre-training to focus on relevant information and learn meaningful representations. ATM achieves 21.0% WER on mismatched AMI SDM set with guided masking and a 20.7% WER is obtained by including loss scaling (ATM+S). We empirically show that ATM is more robust to changes in masking percentage compared to random masking. as typically used in MSM. Our results substantiate the importance of learning from high confident frames by attaining improvements across multiple evaluation sets. An important aspect of ATM approach is its flexibility to incorporate into any MSM pretraining techniques and ATM+S can also be easily adopted into self-supervised pre-training methods. In our future work, we wish to apply ATM over pretraining data containing data from multiple domains (Hsu et al., 2021a; Likhomanenko et al., 2020) to achieve further improvements. We also consider two future enhancements to ATM: (1) Joint training of the scorer model with MSM model by simultaneous training on supervised and unsupervised data. (2) Perform active learning by sharing the parameters of MSM with the scorer once the MSM is well trained.

## ETHICS STATEMENT

While we are unaware of any specific bias in Ask2Mask, it is possible that bias within the Speech-Stew and LibriLight training corpora themselves may introduce bias to the resulting ASR model. Ask2Mask uses an ASR model trained on labeled data as a scoring model identify regions for masking; any bias in this scoring model (trained on LibriSpeech-100 or AMI) could also impact the fine-tuning process, leading to bias in the final model. Without being aware of specific biasing effects, it is unclear how, if at all, biases in the scoring model and downstream ASR model may interact with each other.

Automatic Speech Recognition (ASR) can exacerbate security and privacy issues by facilitating the search and analysis of speech, though in and of itself ASR does not pose a security nor privacy issue.

## REPRODUCIBILITY STATEMENT

All of the corpora used for 1) training the scorer model, and 2) pretraining and finetune the ASR models, as well as 3) all evaluations are publicly available and use standard published partitions. We aim to describe Ask2Mask and Ask2Mask with scaling comprehensively in the text to facilitate reproducibility. Implementation of major components, Conformer RNN-T models, wav2vec 2.0 and CTC training are available within in public, open-source libraries including Nvidia-nemo (link) and ESPNet (link). This supports the reproducibility of Ask2Mask and the results reported in this paper within multiple frameworks. Additional details to promote reproducibility are provided in the Appendix.

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

# A   APPENDIX

## A.1   DETERMINE THE BEST PT AND FT CHECKPOINT (CKPT.)

Selecting the best PT ckpt. to perform the fine-tuning is cumbersome as it involves finding the best PT ckpt. based on the validation accuracy on quantized targets and also run fine-tuning experiments on multiple PT ckpts. Figure 3 shows that ATM starts to show improved performance after 250k PT ckpt. On contrary to our belief that ATM might require less FT, both baseline and ATM requires same number of FT iterations to achieve the best performance.

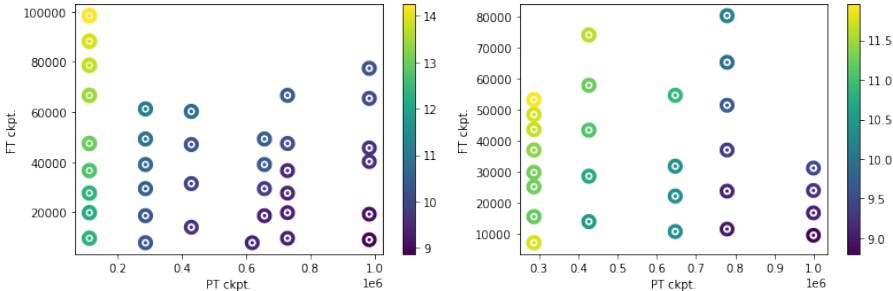

Figure 3: 2D plot of the %WER on IHM-eval across PT and FT ckpts. using baseline (left) and ATM (right) models. The FT is done using AMI.

## A.2   ANALYSIS OF SCORER BASED ON %WER ON PT DATA (1 HOUR OF LIBRI-LIGHT )

We further analyse the impact of the recognition performance of the scorer on ATM, by choosing three categories of scorers such as bad, better and best. The 10 hours of supervised libri-light (LL-10h) data is used to evaluate the performance of these scorers and the results are as follows: Initial model checkpoint (ckpt.). LS-scorer attains 51.8% WER and AMI-scorer achieves 72.7% at this

ckpt respectively. Intermediate model ckpt. Evaluating LS-scorer at this ckpt., achieves 45.8% and AMI-scorer achieves 57.1% WER. Final model ckpt. chosen after convergence. The LS-scorer gains 35.5% WER and AMI-scorer attains 46.8% by evaluating at this ckpt. Figure 4 shows a more

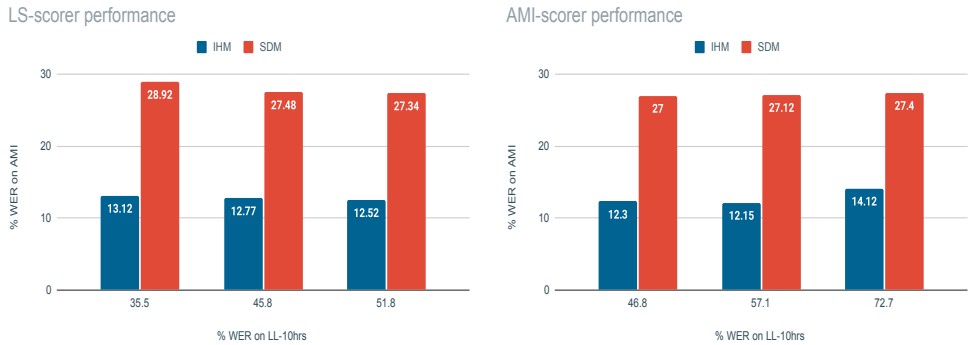

Figure 4: Comparing the behavior of ATM using LS-scorer and AMI-scorer having different %WER on 1 hour of Libri-light (LL-1hr) with AMI evaluation sets)

challenging scenario by testing the ability of ATM using scorers from different domains with varying range of performances. A LS-scorer with 51.8%WER on LL-1hr performs well on both IHM-eval and SDM-eval, while the AMI-scorer with 46.8% WER slightly improves over the above mentioned LS-scorer. The key aspect in selecting better scorer is to choose ckpt. either from the category which achieves %WER within the range of 50-60 on the small subset of pre-training data.

### A.3 ANALYSIS OF ATM ON LIBRISPEECH

Our ATM analysis is also done on w2v-BERT-L to validate the ATM on LS-100 and is present in table 7. ATM attained 4.3% and rel. imp. on test-other over the baseline model. Apart from that there is a slight change in test set which can be treated as noise. On both dev-other and test we did not find any improvement and instead there was slight degradation in performance.

Table 7: %WER of ATM by varying the frames selection based on high, low and mixed confidence (conf.) scores using w2v-BERT-L on all evaluation sets on Librispeech.

| Model | Type | PT-LL, FT-LS100 | | | |
|---|---|---|---|---|---|
| | | dev | dev-other | test | test-other |
| w2v-BERT-L | Baseline | 3.78 | **8.86** | **3.85** | 9.32 |
| | High-conf. | **3.71** | 8.97 | 3.89 | **8.92** |
| ATM | Low-conf. | 3.98 | 9.59 | 4.15 | 9.76 |
| | Mixed-conf. | 3.87 | 9.25 | 3.98 | 9.42 |

To further evaluate the impact of increasing the model parameters from "L" size to "XL" size, we FT on LS-100 using MSM models with XL size and the results are in table 8. We did not find any consistency in the performance across the evaluation sets using any of the MSM architectures. Slight gains are observed on test or dev or dev-other using w2v2-conformer-XL. Once the baseline in w2v-BERT-XL gets better, ATM did not achieve gains on test-other. This scenario can be explained due to effectiveness of MSM pre-training under matched condition and can perform well without any ncessary data selection approach.

### A.4 ATM+S ANALYSIS ON VALIDATION DATA DURING PT

The effect of ATM and ATM+S is analysed by plotting the validation scores on dev-other during pre-training. The first plot in figure 5 shows that the contrastive loss improves over the baseline with the aid of ATM and is further enhanced with ATM+S. The second plot shows the number of unique

Table 8: Performance comparison of different MSM architectures with and without applying ATM on all evaluation sets on Librispeech.

| Model | Type | PT-LL, FT-LS100 | | | |
|---|---|---|---|---|---|
| | | dev | dev-other | test | test-other |
| w2v2-Conformer-XL | Baseline | 2.5 | 4.7 | 2.6 | 4.9 |
| | ATM | 2.4 | 4.6 | 2.5 | 5.0 |
| HuBERT-Conformer-XL | Baseline | 2.5 | 4.7 | 2.6 | 5.0 |
| | ATM | 2.5 | 4.6 | 2.5 | 5.0 |
| w2v-BERT-XL | Baseline | 2.4 | 4.4 | 2.5 | **4.6** |
| | ATM | **2.3** | **4.4** | **2.4** | 4.7 |

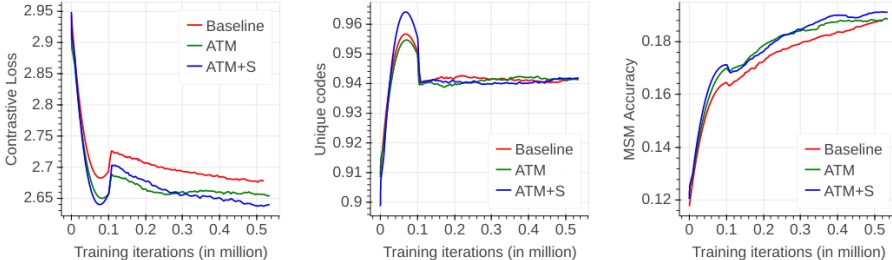

Figure 5: Validating baseline, ATM and ATM+S using contrastive loss, number of unique codes used from the codebook and the MSM accuracy on dev-other during pre-training. A small bump is observed at around 0.1 million training iteration due to the change in learning rate and is ignored during analysis.

codes used from the quantizer codebook. Analysing this plot helps us to understand if the validation loss or accuracy is improved by just using less % of unique codes which will affect the performance at FT. Among the 1024 codes, 94%-95% are used by both ATM and ATM+S. This is similar to the % unique codes used by the baseline model and confirms that improvement of ATM and ATM+S is not by choosing smaller set of unique codes. The third plot shows that the MSM accuracy of ATM and ATM+S improves over baseline model. ATM+S shows that re-weighting each utterance is complementary to the ATM.

## A.5 COMPARISON BETWEEN FRAME-LEVEL AND UTTERANCE-LEVEL LOSS SCALING

ATM+S performs MSM loss scaling using the utterance-level confidences which performs focus on each utterance at a coarse level. We also experimented with scaling with frame-level confidence scores. Our experiments showed that scaling all utterances with frame-level confidence hurts the model performance. To solve this issue, we randomly selected utterances which participate in frame scaling. Scaling 10% utterances resulted in better performance and the results are shown in Table 9.

Table 9: %WER on Librispeech evaluation sets using ATM with utterance scaling and frame scaling. The frame scaling is analysed with choosing the best percentage of utterances that participate in scaling.

| Model | Type | PT-LL, FT-LS100 | | | |
|---|---|---|---|---|---|
| | | dev | dev-other | test | test-other |
| w2v-BERT-L | Baseline | 3.78 | 8.86 | 3.85 | 9.32 |
| | None | 3.71 | 8.97 | 3.89 | **8.92** |
| | Utterance-level | **3.64** | **8.79** | 3.95 | 8.95 |
| ATM | Frame-level-10% | 3.73 | 8.97 | **3.84** | 9.06 |
| | Frame-level-50% | 4.01 | 9.35 | 4.14 | 9.54 |
| | Frame-level-100% | 4.23 | 9.77 | 4.89 | 9.97 |

### A.6 RESULTS ON COMMONVOICE WITH PUNCTUATION NORMALIZATION

Some previous work, including Likhomanenko et al. (2020), report Commonvoice results by scoring with a normalization process that removes punctuation. For comparison, we do the same in Table 11.

Table 10: %WER on Commonvoice using models FT with speechstew.

| Model | Without norm. | With norm. |
|---|---|---|
| Speechstew (Chan et al. (2021)) | 12.1 | 9.7 |
| w2v2-Conformer-XL (Chan et al. (2021)) | 11.5 | 9.1 |
| w2v-BERT-XL | 11.2 | 9.3 |
| w2v-BERT-XL + ATM | 10.8 | 9.2 |
| w2v-BERT-XL + ATM+S | 10.7 | 9.1 |

### A.7 BLOCK DIAGRAMMATIC VIEW OF MSM ARCHITECTURES

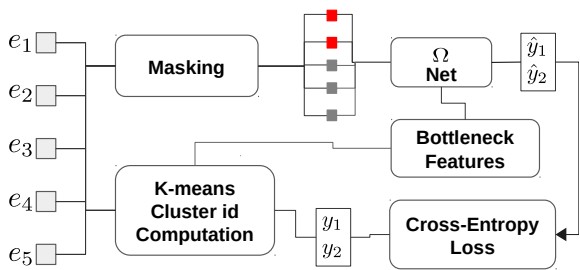

Figure 6: Working procedure of HuBERT-conformer model as described in section 2.3. The k-means cluster ids act as labels and they are refined using the bottleneck features extracted from the context network itself.

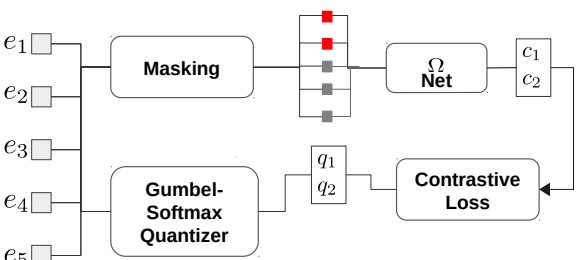

Figure 7: Working procedure of Wav2vec2-conformer model as described in section 2.3. The encoded representations are masked and passed to context network $\Omega$ and the resulting output $c_j$ is learnt to be closer to quantized output

### A.8 STATISTICAL SIGNIFICANCE ANALYSIS ON %WER PERFORMANCE FOR MULTIPLE EVALUATION SETS

The table 4 results are on Librispeech and obtaining 0.1% improvement in Librispeech testsets is statistically signficant. For instance, the dev-clean test set contains 54402 words and 0.1% gains denotes a recovery of 54 words. Also, the recent works on self-supervised training such as HuBERT shows improvement between wav2vec2-Large and HuBERT-Large only on dev-clean with 0.1% gain in Table 3.

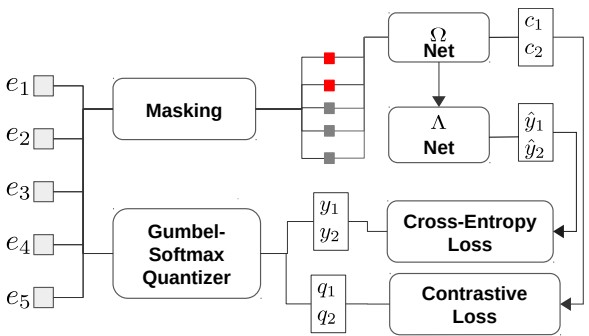

Figure 8: Working procedure of W2V-BERT model as described in section 2.3. Cross-entropy loss is computed between predictions of context network $\Lambda$ and the quantized labels $y_j$. Contrastive loss is computed in parallel as in wav2vec2.

Table 11: %WER on Commonvoice using models FT with speechstew.

| Dataset | Abs. %WER Imp. | # Total words | # Words recovered ($\approx$) |
|---|---|---|---|
| dev-clean | 0.1 | 54402 | 54 |
| test-clean | 0.1 | 52576 | 52 |
| AMI-IHM | 0.1 | 89635 | 89 |

