# OpenReview forum: "Ask2Mask: Guided Data Selection for Masked Speech Modeling"
_ICLR.cc/2022/Conference — ICLR 2022 Submitted_

### Official Review · Reviewer_EBBL · 2021-10-20

**Correctness:** 3
**Technical Novelty And Significance:** 3
**Empirical Novelty And Significance:** 3
**Recommendation:** 5
**Confidence:** 5

**Main Review:**

Strengths
- paper introduces confidence scores into masked modelling
- paper shows that even simplest of schemes can lead to small-to-medium gains
- comprehensive performance comparison across many speech data sets
- large-scale, non-trivial, models and data sizes

Weaknesses
- limited technical novelty
- technical quality of paper (numerous typos (eq 2, ), undefined terms ({\bm E}^{rm}), sloppy notation (1,...,k-1 does not equal K), confusing notation (eq 5)); section 3.1 is meant to be the key section but you are making a very limited attempt to be clear not just for speech but also more general audiences
- limited discussion about the use of confidence scores in ASR
- lack of any discussion or investigation regarding suitability of softmax probabilities as surrogates of confidence scores
- your claim about gains up to 11.6% seems to apply to CV task if you compare 12.1% of SpeechStew and your ATM+S approach. If this is true your statement in the abstract is misleading at best. Please focus on comparing unweighted and weighted schemes. Please make it clear when you are comparing your best number and someone else's best numbers.

**Summary Of The Paper:**

This paper describes a weighted masking scheme for learning speech representations. Weights in this scheme are surrogates of confidence scores obtained from an external ASR system. The paper also describes how utterance-level confidence scores can be incorporate to down-weight contribution from utterance that are likely to be problematic for an ASR system.

The paper makes use of Librispeech dataset for learning representations and Librispeech and AMI datasets for training external ASR systems. The paper evaluates both schemes on Librispeech, AMI, and some other datasets. Experimental results show that the proposed schemes offer small-to-medium gains over the unweighted reference masking scheme.



**Summary Of The Review:**

This paper explores the use of weighted masking for learning speech representation. The proposed scheme relies on confidence scores obtained from external ASR systems to decide which speech segments should and should not be used for masked modelling. An extensive empirical investigation shows that weighted masking is a promising direction for learning accurate speech representations.

Although the technical novelty is limited, this paper introduces weighted masking by means of confidence scores (very popular subject) into speech representation learning area. Unfortunately, this paper currently contains a number of technical issues (see above). Furthermore it offers a limited insight about the quality of confidence scores used and sensitivity of the proposed scheme to this important in practical applications factor (e.g. limited resource languages where WERs of typical external ASR systems range wildly).

---

> ### Author Response · Authors · 2021-11-12
> **Response to reviewer EBBL**
>
> First, we thank you for all your valuable comments.
>
> limited technical novelty
>
> -   We have addressed your concerns defined in the weaknesses mentioned. We hope this will help to provide better clarity and highlight the importance of this work.
>
> technical quality of paper (numerous typos (eq 2, ), undefined terms ({\bm E}^{rm}), sloppy notation (1,...,k-1 does not equal K), confusing notation (eq 5)); section 3.1 is meant to be the key section but you are making a very limited attempt to be clear not just for speech but also more general audiences
>
> -   We apologize for the lack of clarity. We have revised Section 3.1 to the following in hopes of addressing this and also added a separate section 3.3 to compare with related works in ASR literature.
>
> - The ATM technique uses probability as a simple form of confidence measure for each frame. Popularized in the speech recognition literature since 2001 [1,2,3,4], ....The analysis of confidence measures for semi-supervision in ASR has been done in [4] and they show that posterior probability acts as a reliable confidence measure for frame, word and sentence based data selection. They also perform an extensive analysis on using the posteriors for hybrid ASR systems. A similar observation has been noted in [3], where the usage of softmax probability directly as a confidence measure has been applied to select relevant data samples during training. Based on this volume of work, we chose to use softmax probability directly as our confidence score. We also experimented with Entropy and exponential scaling or log scaling on softmax probabilities as confidence measure, but it did not fetch any additional advantage over raw probability. Thus, we settled on the simplest approach.
>
> -   [1]  Lamel, Lori & Gauvain, Jean-Luc & Adda, Gilles. (2001). Lightly Supervised Acoustic Model Training. 10.1109/ICASSP.2002.1005880.
>
> -   [2]  de Chaumont Quitry, Félix, et al. "High quality agreement-based semi-supervised training data for acoustic modeling." 2016 IEEE Spoken Language Technology Workshop (SLT). IEEE, 2016.
>
> -   [3] Jonathas Ferreira, Marcele Mendonca, and Paulo SR Diniz. Data selection in neural networks, IEEE Open Journal of Signal Processing, 2021 (<https://ieeexplore.ieee.org/stamp/stamp.jsp?tp=&arnumber=9519166>).
>
> -   [4] Veselý, Karel, Lukás Burget, and Jan Cernocký. "Semi-Supervised DNN Training with Word Selection for ASR." Interspeech. 2017.
>
> limited discussion about the use of confidence scores in ASR
>
> -   Thank you for raising this point. The analysis of confidence measures for semi-supervision in ASR has been done in the paper "Semi supervised DNN training with word selection for ASR" (Veselý and Burget 2017). This work is cited in our paper as it shows that posterior probability acts as a reliable confidence measure for frame/word/sentence based data selection. (Veselý and Burget 2017) does extensive analysis on using the posteriors for hybrid ASR systems. Based on the motivation from this work we chose to use softmax probability directly as our confidence score. This information has been included in an updated draft in the aforementioned mentioned additional paragraph in Section 3.1.
>
> -   We have also included a new Section 3.3 in the paper which discusses confidence scores used in prior ASR work and the motivation behind its usage in Ask2Mask.
>
> lack of any discussion or investigation regarding suitability of softmax probabilities as surrogates of confidence scores
>
> -   The usage of softmax probability directly as a confidence measure has been applied in this work "Jonathas Ferreira, Marcele Mendonca, and Paulo SR Diniz. Data selection in neural networks, IEEE Open Journal of Signal Processing, 2021 (<https://ieeexplore.ieee.org/stamp/stamp.jsp?tp=&arnumber=9519166>).
>
> -   We also experimented with exponential scaling or log scaling on softmax probabilities, but it did not provide any advantage over simple usage of probability. This information has been included in an updated draft in the aforementioned mentioned additional paragraph in Section 3.1.
>
> your claim about gains up to 11.6% seems to apply to CV task if you compare 12.1% of SpeechStew and your ATM+S approach. If this is true your statement in the abstract is misleading at best. Please focus on comparing unweighted and weighted schemes. Please make it clear when you are comparing your best number and someone else's best numbers.
>
> -   We apologize for the lack of specificity in this statement and are grateful for the opportunity to clarify.  We have expanded the abstract to be more specific about the relative improvement over published state of the art results and our internal baselines.
> -   From 1.1 to 11.6% relative improvement over SpeechStew, and 1.8 to 4.5% improvement over our internal baseline.
> -   We are including an additional paragraph in the results section to discuss the results (changes are marked in red).

---

### Official Review · Reviewer_YoKo · 2021-11-02

**Correctness:** 3
**Technical Novelty And Significance:** 3
**Empirical Novelty And Significance:** 2
**Recommendation:** 5
**Confidence:** 4

**Main Review:**

Strengths:

- Experiments are extensive to evaluate different aspects of the proposal (amount of masking vs. WER, type of scorer vs WER, effect of ATM approach vs. ATM+scaling approach)
- The method is relatively easy to implement with existing open source tools.

Weaknesses:

Notation of the equations needs some updates, for example:
- Eq. 2. $L$ is used both as an index and as a set of codebook indices
- Eq. 5. The denominator should use a summation index different than $t$

The novelty is somewhat limited. Having a weighting mechanism instead of uniform distribution for masking is relatively straightforward extension of the existing masked based modeling approaches.

Some details are missing. Please refer to the questions below:
1) Introduction: “The scoring model is necessarily a speech recognition model trained on small amount of data” -> it could have been a large model on a known larger dataset. Would a better scorer help with ATM?
2) “ignoring utterances with low-confidence” -> Not exactly true, to the reviewer's understanding, confidence level is just a scaling factor. Or do the authors put a threshold on scores?
3) In the introduction of Section 5, which part of AMI is used as the training subset? Without mentioning this detail, it is hard to see why it can show the domain generalization aspect. Since the experiments usually mention LS fine-tune -> test on LS test, and AMI fine-tune and test on AMI test set, the generalization claim is not very clear.
4) Details of fine-tuning is missing. Do the authors use, for instance, w2v-BERT-L + ATM based embeddings and train a new ASR model? Or do the authors add a new output layer to map input speech to word-piece units?
5) In Table 4 and its explanation, do the authors refer to match and mismatch based on the dataset used in pre-training and fine-tuning?

Minor typos: Introduction, first paragraph: pesudo -> pseudo;
Confidences cores -> confidence scores

Please add reference number to CHiME-6 dataset in Section 4.1.

**Summary Of The Paper:**

This study investigates the use of an external frame-synchronous CTC-based ASR system to get confidence scores for frames in masked speech modeling which is called AskToMask (ATM). These confidence scores are used in two ways:
1) as a distribution to determine the segment to mask (higher confidence means higher probability),
2) the utterance level average of the per-frame confidence scores is used as a scaling factor of the per-utterance loss.

The experiments perform unsupervised pre-training using these two proposed methods and then performs supervised fine-tuning for the final ASR task. Experiments on Librispeech and AMI show that confidence score-based selection of masking improves the final WER as compared to using uniform distribution in the mask selection (the first approach versus baseline). The second approach provides marginal gains on top of the first approach.

**Summary Of The Review:**

The paper uses ASR based scores to mask high-confident regions in masked speech modeling. Using this weighting idea is not very novel even though it may be the first implementation in the speech domain. The main strength of the paper is that it provides extensive comparison for different aspects of the approach. The writing may benefit from some notational fixes and addition of some implementation details as discussed in the detailed review. From the discussion above, it can be seen that there are more weaknesses as compared to the strengths.

---

> ### Author Response · Authors · 2021-11-12
> **Response to reviewer YoKo**
>
> We thank the reviewer for understanding our technique and their valuable comments
>
> Notation of the equations needs some updates,
> -   Thank you for pointing these out to us. We have incorporated the above corrections into the paper draft.
>
> The novelty is somewhat limited. Having a weighting mechanism instead of uniform distribution for masking is relatively straightforward extension of the existing masked based modeling approaches.
> -   Thank you for your clear suggestions. In this work, we implement this straightforward and simple idea to improve ASR as it was not tried in self-supervision research in ASR. Our idea is that this simple technique will be easy to implement and the improvements are achievable with less changes to the architecture or loss function.
> -   The weighted scheme using confidence scores has been successful in semi-supervised, unsupervised and active learning approaches in ASR for HMM-based and neural network based systems [shorturl.at/hvwAR,shorturl.at/hxKMS].].  However, to the best of our knowledge these confidence scores have not been utilized in state-of-the-art complex neural architectures with  a large number of parameters.  We believe that the  novelty factor lies in bringing the conventional usage of confidence scores into self-supervised training successfully. The importance of this work is that a simple, easy-to-implement technique can fetch gains without increasing training complexity in a fairly complex self-supervision framework.  Our approach is focused on improving ASR across multiple datasets and our hypothesis is that our simple approach is quite effective.
> -   The baseline w2v-BERT-XL model used in this work attains better results over the published w2v2-conformer-XL and Speechstew results. In CommonVoice and CHiME-6, the baseline attained 7.4\% and 2.9\% relative improvement over Speechstew respectively. Our approach improves over our own baseline with a consistent improvement across all range of mismatched domains by attaining upto 4.4% relative improvement over the baseline.
>
> Introduction: "The scoring model is necessarily a speech recognition model trained on small amount of data" -> it could have been a large model on a known larger dataset. Would a better scorer help with ATM?
> -   Based on premilinary empirical analysis, we found that the difference is very marginal and hence our focus was less on scorers' performance.
>
> "ignoring utterances with low-confidence" -> ...... do the authors put a threshold on scores?
> -   Thank you for pointing it out. Yes your understanding is correct, We don't strictly "ignore" the utterances with low-confidence by instead down scale them. We have incorporated this correction in the paper draft as "down-scaling the utterances with low-confidences".
>
> In the introduction of Section 5, which part of AMI is used as the training subset?  .... ... the generalization claim is not very clear.
> -   AMI corpus is a 100 hour dataset which includes speech from close-talking microphone and far-field microphones recorded during meetings. This is different from the Libri-light corpus which are audio books recorded with a single microphone by reading the books.  The idea behind this work is to boost the performance when there is a mismatch between data used during PT and FT domain. Here the generalization capability is considered to be better when the model improves regardless of the mismatch between PT and FT. This can be seen in table 3 and 5, where we pretrain with Libri-light and fine-tune with AMI using ATM and compare with baseline model.
>
> Details of fine-tuning is missing. --- train a new ASR model? Or do the authors ... word-piece units?
> -   In Section 4.3 we have  defined the fine-tuning configuration and we have also incorporated additional text to describe the finetuning procedure in Section 4.3 of the new paper draft.  "Here the pretrained network denotes the context network and is used as the encoder for the final model.  A decoder with two layers is added and  fine-tuning is done by re-training both the encoder and decoder until convergence."
>
> In Table 4 and its explanation, do the authors refer to match and mismatch ... ?
> - Yes, that’s correct. We have added the following lines to the caption for Table 4 for further clarification:
> "%WER obtained by FT with LS-960 using   baseline, w2v-BERT-XL, ATM and ATM+S models. The results show the impact of our proposed approach on matched conditions. Librispeech and  Libri-light corpora are matched in that they are speech from audio books."
>
> Minor typos:
> -   Thank you for pointing out these edits, we have incorporated it in the paper.
>
> From the discussion above, it can be seen that there are more weaknesses as compared to the strengths.
> -   We have done our best, given the time constraints, to address these weaknesses mentioned in this review. We hope this provides clarity and highlights the importance of this work.

---

### Official Review · Reviewer_fRrU · 2021-11-03

**Correctness:** 4
**Technical Novelty And Significance:** 2
**Empirical Novelty And Significance:** 2
**Recommendation:** 6
**Confidence:** 5

**Main Review:**

I personally think that authors have a good point here, how to do better mask selection in self-supervised learning. Obviously, not all speech frames are
of equal importance. Studies, such as in automatic speaker recognition have been made, where it was clearly found out that different phones carried
different amount of speaker recognition cues than others. In ASR, you can take non-speech region vs vowel and you get a big difference in importance.

However, the way it was done here, I believe is not correct. First of all, the idea of self-supervised learning is to learn task independent features that
can be then used for any task. So, then in principle, the learning signal should be something that does not have any one task in mind. But okay, lets say
your intention is to do only ASR and your chosen model, of course then you can proceed as described in the present paper. But then your need to compare
with other ASR models that use _similar_ amount (and the same) labeled training data. Which in your case is the data used in the scoring model. So SoTA
end-to-end mehtods need to be used (and ESPnet is a good way to start in looking for implementations https://github.com/espnet/espnet).

And finally, if you look at the Table 6, the difference in WER betwene proposed, that uses extra data, vs baseline is not too great. Only in CHIME we see
some real improvement.

Minor comments:

- L_div is not explained in more detail.
- Equation (3) is not needed.
- This sentence is very suspicious: "The scorer’s training data is chosen such that it is matches the target data condition". Wouldn't it be better
that the whole learned model is as general as possible? Could you elaborate on this?
- In Table 2, could you add the metric (WER) in the caption.
- Please replace string "%WER" with "WER (%)". Also in the same vein it would be important to make all captions self-contained, so that
reader should be able to understand the Table and Figure without referencing to the the main text.

I have read the authors rebuttal and discussion and due to sufficient answer to my concerns I have raised the score.

**Summary Of The Paper:**

In this paper, authors proposed to use an external scorer to weight the frames to be masked for the MSM loss. Idea is interesting as not all speech frames are of equal importance.

**Summary Of The Review:**

Authors add more labeled training data, as an external scorer, and still results are significantly better. Authors need to compare against SoTA models where all models see exactly the same data.

---

> ### Author Response · Authors · 2021-11-12
> **Response to reviewer fRrU**
>
> Thank you very much for providing your thoughts and suggestions. We have addressed the reviewers questions below:
>
> However, the way it was done here, ..... . But okay, lets say your intention is to do only ASR and your chosen model, of course then you can proceed as described in the present paper.
>
> -   Thank you for mentioning your train of thought and perspective on this work and self-supervision in general. Our motivation to this approach is also to improve ASR for harder tasks without affecting the performance on other datasets. Our results on table 6 show that we not only improve on CHiME-6 and AMI but also on the CommonVoice test set which is closer to pre-training data domain.
>
> But then you need to compare with other ASR models .... Which in your case is the data used in the scoring model.
>
> -   We apologize that it was not clear from the paper that we have not used extra supervised data in experiments in this paper. In most cases, the scorer model is trained on a subset of the fine-tuned data.
> -   While we appreciate the reviewer's perspective on self-supervised pretraining, there are a number of approaches in the literature that do not conform with this perspective.  There is, to be sure, unique value in task-agnostic pretraining tasks.  However, what Ask2Mask as well as other approaches like wav2vec2-S (shorturl.at/lEX18) show, there is value to self-supervised criteria for task-dependent pretraining. We would hope that the reviewer can expand their definition of the research value of self-supervision and pretraining to include not only this paper, but other efforts in this direction (shorturl.at/lEX18 and shorturl.at/tAGS6)
> -   The labeled data we used to build the scorer model is the same data which is used for fine tuning. For instance in Table 4: the scorer is trained with 100 hours of supervised Librispeech data and the fine-tuning is done with 960 hours of Librispeech. Thus we have not used any  additional supervision when compared to the baseline.
> -   In Table 6, SpeechStew is finetuned with multiple datasets, including 960 hours of supervised Librispeech data. Our scorer's training data (100 hours of Librispeech) is a small subset of the SpeechStew data. We therefore believe that the comparisons with the published results are fair and particularly good given that we do not use all of the  additional data used by the SoTA baseline. This highlights the fact that our gains are not related to the amount or domain of the  supervised data (100 hours) used in pre-training, instead they appear  related  to the selection of the highly confident  frames for masking (Table 1).
> -   Thus we humbly suggest that the results we have compared with SpeechStew and other published works are fair comparisons as we only used the same supervised and unsupervised data as the cited works.
>
> And finally, if you look at the Table 6, the difference in WER between proposed, that uses extra data, vs baseline is not too great. Only in CHIME we see some real improvement.
>
> -   Based on our explanation to the previous comment, it is clear the we have not used additional supervision and compared the results with relatively lesser supervision. Hence, we believe our results show the importance of our approach across multiple datasets without loss of generality. We would also like to emphasize that our results are improved over the published SoTA results in literature on multiple datasets. While some of these improvements are smaller than others, considering the amount of prior attention that these tasks have received, 2% relative reduction of error is, with respect, not a modest improvement. We believe the improvements over those WERs are by no means incremental especially given the low WERs for many of these tasks
>
> Minor comments:
>
> -   Thank you for pointing out the minor mistakes and we have corrected them in the paper draft.
>
> This sentence is very suspicious: "The scorer's training data is chosen such that it is matches the target data condition". ... Could you elaborate on this?
>
> -   We have included the following lines in section 3.1 to clarify: "Our initial intuition was to choose the scorer's training data to match the target data distribution. However, our empirical analysis in (cf. Section 5.3) shows that the performance is agnostic to the data used to train the scorer model."
>
> -   Our results in Table 2 shows that regardless of the training data used by the scorer our results are comparable.
>
> Authors add more labeled training data ... compare against SoTA models where all models see exactly the same data.
>
> -   We apologize that this point wasn't more obvious in the submitted draft.  We have not added additional training data when compared to published state of the art models and results.  While Ask2Mask uses the data differently (reusing some data both in training the scorer model and in fine-tuning the ASR model), the topline experiments which we compare to the state of the art models use identical data.

---

### Official Review · Reviewer_UPo9 · 2021-11-04

**Correctness:** 3
**Technical Novelty And Significance:** 2
**Empirical Novelty And Significance:** 2
**Recommendation:** 3
**Confidence:** 4

**Main Review:**

Improving training effectiveness through better masking is an important research direction for speech representation learning.

Strengths:
The paper presents a robust experimental setup with three SOTA models (W2V, HuBERT, W2V-BERT) evaluated in two sizes L/XL on multiple datasets (LS, AMI, CHIME-6, Tedlium, and Commonvoice.

Weaknesses:
There are three major points here:
1) The paper presents an incremental improvement over the baseline systems with minor novelty.
2) The proposed approach relies on access to labeled data during the pre-training stage, which departs from the standard setup for self-supervised learning. The pre-training stage in self-supervised learning is task agnostic to open the door for utilizing the learned representations in a wide array of downstream tasks.
3) The observed gains are modest using the proposed approach. With 100h of labeled data, semi-supervised methods (either from random weights or initialized from self-supervised models) would bring much lower WERs.

General comments:
Equation 5 encourage sampling from places with high confidence, which may lead to oversampling silence and vowel segments. How would you encourage the selection of diverse positive samples?
The primary motivation for filtering data in semi-supervised learning is to remove noisy inputs and ones with bad teacher labels/confidence. A similar goal can be achieved in self-supervised learning using voice activity detection (VAD) to remove long silences and other signal processing methods. Dealing with data problems in self-supervised learning has also been studied in (https://hal.archives-ouvertes.fr/hal-03070411/document)
One way to benefit from a supervised teacher is to benefit from the teacher segmentations and labels, not only confidence scores, for masking and negative sampling as done here (https://arxiv.org/pdf/2103.05149.pdf)


**Summary Of The Paper:**

Many self-supervised speech representation learning methods use masked prediction at their core. This paper proposes Ask2Mask (ATM) approach for informed masking during learning through a supervised teacher model that provides frame-level posteriors probabilities of linguistic output units. The frame-posterior probabilities extracted from the supervised teacher model are also used as utterance weights to favor ones with high confidence.

**Summary Of The Review:**

The paper presents an approach for improving self-supervised learning approaches using a supervised teacher model. The problem is important, but the proposed method and the results aren't convincing.

---

> ### Author Response · Authors · 2021-11-12
> **Response to reviewer UPo9**
>
> We thank the reviewer for the valuable comments and here we have addressed as follows:
> The paper presents an incremental improvement .....
>
> -   Confidence scores have been successful in unsupervised ASR based on HMM-based and neural network based systems [shorturl.at/hvwAR,shorturl.at/hxKMS].  However, to the best of our knowledge these confidence scores have not been utilized in SoTA complex neural architectures and self-supervised training successfully. The importance of this work is that its easy-to-implement technique can fetch gains without increasing training complexity. Our approach is focused on improving ASR across multiple datasets and its impact can be seen in the wins we have shown over the SoTA baselines.
>
> The proposed approach relies on access to labeled data during the pre-training stage,...
> -   Approaches that incorporate external knowledge during pre-training are quite common in ASR. For instance,  Wav2vec2-S (shorturl.at/lEX18) and  Hubert uses a "chenone" recognizer to hypothesize targets for self-supervised PT as a contrast to fully unsupervised learning of targets..
>
> The proposed approach.... labeled data ...  task agnostic to open the door for utilizing the learned representations in a wide array of downstream tasks.
>
> -   While we appreciate the reviewer's perspective on self-supervised PT, there are a number of approaches such as wav2vec2-S and shorturl.at/tAGS6  in the literature use labeled data.  The labeled data we used to build the scorer model is the same data which is used for fine tuning.  For example in Table 6, SpeechStew is finetuned with multiple datasets, including 960 hrs of supervised LS data. Our scorer's training data (100 hrs of LS) is a small subset of the SpeechStew data. We therefore believe that the comparisons with the published results are fair and particularly good given that we do not use all of the  additional data used by the state-of-the-art baseline. This highlights the fact that our gains are not related to the amount or domain of the  supervised data (100 hrs) used in pre-training, instead they appear  related  to the selection of the highly confident  frames for masking (Table 1).  Our work is targeting research specifically towards ASR that is more robust across a wide range of domains.
>
> -   We are unaware of any published result to support the following claim: "With 100h of labeled data, semi-supervised methods (either from random weights or initialized from self-supervised models) would bring much lower WERs." We would be extremely grateful if you would include a reference for this as it would clearly strengthen our work.
>
> Equation 5 encourages .... diverse positive samples?
>
> -   The diversity among samples is the primary concern in masking. Our work, uses a span of 10 frames starting from the sampled index. In addition to this the input segments are randomly selected on-the-fly from the Librilight data as described in speechstew paper. In Section 3.1, we describe how we inject some stochasticity into the selection. These two factors preserve the diversity among samples without training redundantly.
>
> The primary motivation for filtering data... (VAD) to remove long silences...
>
> -   Thanks for highlighting the motivation and we were aware of the cited paper.  Its conclusion states that " The cumulative effect of these three factors adds up to a drop of 30% relative ABX score, even though the Libri-light dataset is itself relatively clean (home made audio books). For instance, most of the recording time is devoted to speech, and the non-speech parts are minimal (8.8%); this may be why VAD filtering is having such a small effect". In addition to this prior work using LIbrilight pre-training using wav2vec2 it has been shown that applying VAD did not yield substantial performance gains but instead lead to degradation in performance. (cf. table 4 in <https://arxiv.org/pdf/2010.10504v1.pdf>). We are using the same data (Libri-light) as this paper during pre-training, since we are seeing larger relative wins than were reported using VAD, we have some confidence that the scorer model confidence is more informative than silence/non-speech selection.
>
> One way to benefit from a supervised teacher is ....  as done here (<https://arxiv.org/pdf/2103.05149.pdf>).
> -  We thank the reviewers for pointing out the work to exploit supervised teachers. Our approach is complementary to the CSL and the difference between both approaches is that we don't rely on the pseudo-labels and instead only on the frame-level confidence scores provided by the teacher which allows a possible extension of combining these two techniques. We also would like to emphasize that the valid addition to this, our work also provides utterance-level data selection using ATM+S which can benefit any semi-supervised or self-supervised loss objective without orienting to contrastive loss only as in CSL. We will include this in the related work as an alternative way of using the teacher model.

---

### Author Response · Authors · 2021-11-12
**Brief comment to all reviewers. The reviewers suggestions are incorporated into the paper draft and are highlighted in red colour**

The authors would like to strongly emphasize the following:
 -  While the scorer model is trained with supervised data, the data used in its training is (in almost all reported experiments) a subset of supervised data used in fine-tuning the ASR model. We have also shown that there is limited impact of the domain of the data used to train the scorer model for the scenarios where it is different from that data used to fine-tune the ASR model.
 -  The gains over the existing benchmark results are not trivial. These tasks have been well studied in the field for many years. Moving the needle over the state-of-the-art, across a range of test sets is uncommon. E.g. Figure 1 in https://arxiv.org/pdf/2010.10504.pdf describing progress on LibriSpeech.
 - We didn't use a tool such as ESPNet, because although it's awesome, it still has a substantial gap with the performance of our baseline system cf. “Recent Developments on Espnet Toolkit Boosted By Conformer“ (https://ieeexplore.ieee.org/document/9414858). For example, in Table 4 from this paper, on TEDLIUM3 ESPnet conformer gets 7.6% WER while our conformer baseline attains 5.7% WER and with pretraining this is improved to 5.3% WER. Our aim is to compare our results with the best published numbers in literature regardless of the toolkit.  We would also highlight the simplicity of the proposed model makes it very available for integration with ESPNet and other open source toolkits.
- A simple, elegant and robust confidence metric, easy to implement in any toolkit, pushes the state-of-the-art in well-benchmarked corpora in the speech recognition community.

---

### Decision · Program_Chairs · 2022-01-20

**Decision:**

Reject

**Comment:**

This paper presents an approach that uses ASR-based scores to guide masking high-confident blocks for speech representation learning. As most of the reviewers mentioned, it is an incremental improvement over baseline systems with limited novelty. About the use of confidence scores which is a key factor of the method, it lacks enough discussion on its quality and sensitivity.